# Impact of Pre-Diagnostic Risk Factors on Short- and Long-Term Ovarian Cancer Survival Trajectories: A Longitudinal Observational Study

**DOI:** 10.3390/cancers16050972

**Published:** 2024-02-28

**Authors:** Shana J. Kim, Shelley S. Tworoger, Barry P. Rosen, John R. McLaughlin, Harvey A. Risch, Steven A. Narod, Joanne Kotsopoulos

**Affiliations:** 1Women’s College Research Institute, Women’s College Hospital, Toronto, ON M5S 1B2, Canada; shana.kim@mail.utoronto.ca (S.J.K.); steven.narod@wchospital.ca (S.A.N.); 2Dalla Lana School of Public Health, University of Toronto, Toronto, ON M5T 3M7, Canada; 3Division of Oncological Sciences, Knight Cancer Institute, Oregon Health and Sciences, Portland, OR 97201, USA; tworoges@ohsu.edu; 4Princess Margaret Cancer Centre, University Health Network, Toronto, ON M5G 2C4, Canada; barry.rosen@beaumont.org; 5Division of Gynecologic Oncology, Department of Obstetrics and Gynecology, University of Toronto, Toronto, ON M5G 1E2, Canada; 6Samuel Lunenfeld Research Institute, Mount Sinai Hospital, Toronto, ON M5G 1X5, Canada; john.mclaughlin@utoronto.ca; 7Public Health Ontario, Toronto, ON M5G 1M1, Canada; 8Department of Chronic Disease Epidemiology, Yale School of Public Health, New Haven, CT 06520, USA; harvey.risch@yale.edu

**Keywords:** ovarian cancer, survival, mortality, risk factors, trajectory

## Abstract

**Simple Summary:**

This study had the unique opportunity to evaluate the impact of pre-diagnostic risk factors on short- and long-term ovarian cancer survival trajectories in the Canadian context. We found that clinical factors such as histology, stage, and treatment are predictors of short- and long-term survival. In addition, pre-diagnostic exposures such as breastfeeding, smoking, and BMI influence long-term survival. These findings suggest that modifiable risk factors differentially impact ovarian cancer survivorship, and researchers can explore potential avenues to improve the outcomes of this highly fatal disease.

**Abstract:**

Tumor- and treatment-related factors are established predictors of ovarian cancer survival. New studies suggest a differential impact of exposures on ovarian cancer survival trajectories (i.e., rapidly fatal to long-term disease). This study examined the impact of pre-diagnostic risk factors on short- and long-term ovarian cancer survival trajectories in the Canadian context. This population-based longitudinal observational study included women diagnosed with invasive epithelial ovarian cancer from 1995 to 2004 in Ontario. Data were obtained from medical records, interviews, and the provincial cancer registry. Extended Cox proportional hazard models estimated the association between risk factors and all-cause and ovarian cancer-specific mortality by survival time intervals (<3 years (i.e., short-term survival), 3 to <6 years, 6 to <10 years, and ≥10 years (i.e., long-term survival)). Among 1421 women, histology, stage, and residual disease were the most important predictors of all-cause mortality in all survival trajectories, particularly for short-term survival. Reproductive and lifestyle factors did not strongly impact short-term overall survival but were associated with long-term overall survival. As such, among long-term survivors, history of breastfeeding significantly decreased the risk of all-cause mortality (HR 0.65; 95% CI 0.46, 0.93; *p* < 0.05), whereas smoking history (HR 1.75; 95% CI 1.27, 2.40; *p* < 0.05) and obesity (HR 1.81; 95% CI 1.24, 2.65; *p* < 0.05) significantly increased the risk of all-cause mortality. The findings were consistent with ovarian cancer-specific mortality. These findings suggest that pre-diagnostic exposures differentially influence survival time following a diagnosis of ovarian cancer.

## 1. Introduction

Ovarian cancer remains the most fatal gynecologic disease in Canadian women, with 5- and 10-year survival rates of 44% and 35%, respectively [1,2]. Age at diagnosis, stage of disease, treatment, and tumor histology continue to be the most important prognostic factors [3,4,5]. In particular, studies have consistently demonstrated the long-term survival effect of cytoreductive surgery to reduce residual disease [6,7]. Whether there is a role of reproductive, hormonal, or other modifiable exposures on ovarian cancer outcomes is of interest. In a 10-year cohort of 1421 women diagnosed with epithelial ovarian cancer in Ontario, Canada, we previously demonstrated that pre-diagnostic lifestyle factors—notably, obesity and smoking—were significant predictors of death, even after accounting for established clinical and pathologic prognostic factors [8].

More recently, studies have begun to characterize the differential impact of risk factors on ovarian cancer survival trajectories by classifying cases into rapidly fatal to long-term disease [9,10]. In an analysis of 35,868 cases from the Surveillance, Epidemiology, and End Results (SEER) program, Peres et al. reported that age at diagnosis, stage, marital status, and race/ethnicity were strongly associated with rapidly fatal disease <3 years following diagnosis, but the effects were attenuated for long-term survival ≥9 years after diagnosis [11]. Most administrative datasets such as SEER do not collect information on known and suspected ovarian cancer risk factors; thus, little is known about the impact of pre-diagnostic epidemiological risk factors on ovarian cancer survival trajectories. The few existing trajectory studies have been conducted in predominantly U.S. populations, including the Nurses’ Health Study and the Ovarian Cancer Cohort Consortium [9,12].

Our team has established a large population-based series of unselected epithelial ovarian cancers diagnosed in Ontario, Canada, with detailed information on pre-diagnostic exposures, pathology information, and vital status [8,13,14]. Given the extended length of follow-up time and unique collection of data sources, this resource provides us the opportunity to conduct a detailed evaluation of clinical, reproductive, and lifestyle risk factors on ovarian cancer survival trajectories in the Canadian context. Thus, we examined the association between various pre-diagnostic exposures and all-cause as well as ovarian cancer-specific mortality by short-term and long-term survival trajectory intervals.

## 2. Materials and Methods

### 2.1. Overview of Study Design

This longitudinal observational study invited a population-based series of unselected patients diagnosed with ovarian cancer in Ontario, Canada, for germline genetic testing. The participant enrollment, data collection, and medical record abstraction has previously been described and is summarized in detail below [8,14]. All participants provided written informed consent, and the study was approved by the institutional review boards of the University of Toronto, Women’s College Hospital, and Yale University.

### 2.2. Study Population

The study population consisted of residents of Ontario between 20 to 79 years old diagnosed with invasive epithelial ovarian cancer between January 1995 and December 1999 (Phase 1) and between January 2002 and December 2004 (Phase 2). Participants were invited to undergo genetic testing for germline mutations in *BRCA1* and *BRCA2*. Women who consented to genetic testing and further data collection were eligible for inclusion. Of 3168 eligible participants, consent and a blood sample for testing were obtained from 1421 (45%) women, all of whom were included in the current study [8,14].

### 2.3. Data Sources and Study Measures

Information was obtained from a variety of sources. The Ontario Cancer Registry (OCR) was used to initially identify ovarian cancer cases and, thereafter, to obtain the updated vital status of each case. The OCR is a provincial registry of all cancers (except non-melanoma skin cancer) diagnosed among Ontario residents and obtains data from hospital admission and discharge records, pathology reports, records from all regional cancer centers in the province, and death certificates from the registrar general [15]. According to the OCR, 90% of ovarian cancer cases in the registry have been microscopically confirmed [16].

A review of medical charts, including pathology records, using a standardized abstraction form extracted key clinical data, including histologic subtype of the tumor, staging, treatment, and extent of residual disease following debulking surgery. Demographic, reproductive, hormonal, and lifestyle factors were collected using a study questionnaire that was conducted via a telephone interview using a standardized script. Detailed information was collected for reproductive factors (i.e., age at menarche, age at menopause, parity, breastfeeding) and hormonal exposures (i.e., oral contraceptive use, hormone therapy, infertility treatment), as well as smoking history, height, and weight (at age 21 and five years prior to diagnosis); family history of breast or ovarian cancer; and endometriosis. To avoid the influence of disease status on body weight, we calculated BMI in units of kg/m^2^ using height and weight five years prior to diagnosis. The cumulative number of ovulatory cycles for each woman was estimated by using the following equations: (1) if premenopausal: ovulatory cycles = 12 × [(current age − age at menarche − years of oral contraceptive use − parity × 0.77 − years of breastfeeding)] and (2) if postmenopausal, current age was replaced with age at menopause (5).

### 2.4. Survival Time Intervals

Vital status or, if deceased, the date and cause of death, for all participants was available through the OCR up to 31 December 2020. Death dates were 99% complete for 2019 and 80% complete for 2020. Cause of cause of death was available up to 31 December 2018. Survival time intervals were estimated as days from the date of ovarian cancer diagnosis to the date of death or the date of last study linkage. To estimate the association between risk factors on various ovarian cancer survival trajectories, distinct survival time intervals were created using a similar approach employed in the study by Peres and colleagues to allow for direct comparisons across the studies [11]. Survival intervals were divided into four periods: (1) <3 years (i.e., short-term survival), (2) 3 to <6 years, (3) 6 to <10 years, and (4) ≥10 years (i.e., long-term survival).

### 2.5. Statistical Analysis

Survival outcomes included all-cause mortality and ovarian cancer-specific mortality. Kaplan–Meier curves were used to display the survival probabilities over time (years) [17]. Extended Cox proportional hazards models with Heaviside functions were used to estimate the hazard ratio (HR) band 95% confidence intervals (CIs) between risk factors and mortality by each survival time interval [18]. Proportional hazard assumptions are not applicable to extended Cox models [18]. Heaviside functions allow for the estimation of a constant HR as a function of time in fixed survival time intervals rather than over the entire follow-up period. In the current analysis, it allowed for the estimation of four HRs corresponding to each of the four survival time intervals comprising short-term to long-term survival [18]. The primary outcome, all-cause mortality, modelled participants from the date of ovarian cancer diagnosis until the date of death or the end of follow-up (31 December 2020). The secondary outcome, ovarian cancer-specific mortality, modelled participants until the date of death from ovarian cancer, censoring for death from any other causes or the end of follow-up (31 December 2020). We performed a left-truncated survival analysis to account for survivorship bias accrued in the time elapsed between the date of diagnosis and the date of ascertainment (i.e., genetic testing) [19]. This method of adjustment has been previously shown to reduce the extent of survivorship bias to <10% in this specific study population [20].

All models were adjusted for age at diagnosis (continuous), histology (serous, mucinous, endometrioid, clear cell, and other [mixed, unspecified epithelial]), and stage (I, II, III, IV). A complete case analysis was undertaken where only subjects without missing data were included in each model. In a subset of women (*n* = 833), we further adjusted for residual disease after primary debulking surgery (yes, no). Findings were stratified by early-stage (I, II) and late-stage (III, IV) disease to further investigate the differential impact of stage on mortality. In a secondary analysis of high-grade serous carcinoma (HGSC) ovarian cancers, we limited the cohort to high-grade (2 or 3) serous cases, and in the case of missing grade, stage III or IV serous cases.

Analyses were performed using SAS version 9.4 (SAS Institute, Cary, NC, USA) and all *p-*values are two-sided.

## 3. Results

### 3.1. Cohort Characteristics and Causes of Death

The demographic and clinical characteristics of the study population, as well as the pathological features of the tumors, are summarized in Table 1. The average length of follow-up time was 11.58 years (SD 8.30; range 0.60 to 26.00), and the average age at diagnosis was 57.15 years (SD 11.61; range 20 to 79). Among the 1421 women included in the current analysis, 241 (17%) survived <3 years (short-term survival), 330 (23%) survived 3 to <6 years, 178 (13%) survived 6 to <10 years, and 672 (47%) survived ≥10 years (long-term survival).

Compared to long-term survivors, short-term survivors were on average older at diagnosis (59.4 vs. 54.7 years), had a history of breastfeeding (62% vs. 58%), never smoked (54% vs. 51%), had higher BMI (25.2 vs. 25.7 kg/m^2^), and carried a germline pathogenic variant (i.e., mutation) in the *BRCA1* or *BRCA2* genes (13% vs. 8%). Short-term survivors were also more likely to be diagnosed with late-stage ovarian cancer (27% vs. 4%) and have residual disease after debulking surgery (91% vs. 39%) compared to long-term survivors.

Over the entire follow-up period, 926 (65%) women died, of which the majority died of ovarian cancer (79%), while 18% died of other causes. Among those who died within 3 years of diagnosis (short-term survivors), 95% died of ovarian cancer. Among the long-term survivors (≥10 years), the majority (56%) died from other causes, including other cancer-related deaths (20%), cardiovascular-related deaths (12%), or respiratory infections (7%), while 31% died of ovarian cancer (31%).

Figure 1 shows the ovarian cancer-specific and all-cause survival probabilities over follow-up time (years). For ovarian cancer-specific survival, the probability of surviving 10 years was 51.2%. After 10 years, the probability of dying from ovarian cancer remained stable at 46.6% until the end of follow-up. In contrast, the probability of dying from any cause continued to decline to 31.6% by the end of follow-up, demonstrating that the risk of dying from ovarian cancer plateaued 10 years after diagnosis while the risk of dying from other causes remained high.

### 3.2. Risk Factors and Mortality by Survival Time Intervals

Table 2 summarizes the risk factor associations with all-cause mortality across the four survival time intervals. In all survival time intervals, women diagnosed at age 70 or older were at the highest risk of dying from any cause compared to those diagnosed between ages 50 and 59 years. The evidence was strongest for long-term survivors (≥10 years) where women who were ≥70 years at diagnosis had 2.93 times the risk of dying compared to those 50 to 59 years of age (HR 2.93; 95% CI 1.93, 4.46; *p* < 0.001), and women who were diagnosed prior to age 50 had a significantly decreased risk of dying compared to those who were 50 to 59 years of age (HR 0.43; 95% CI 0.27, 0.69; *p* < 0.05).

Histology, stage, and residual disease following cytoreductive surgery were strongly associated with survival across the entire follow-up period, with the strongest impact observed among women who died within 3 years of diagnosis. Among short-term survivors, women diagnosed with serous ovarian cancer had an increased risk of mortality compared to those diagnosed with endometrioid ovarian cancer (HR 1.97; 95% CI 1.20, 3.22; *p* < 0.05). Serous histologic subtype was associated with increased mortality risk across all survival time intervals; however, the strength of the association attenuated over time (long-term survival HR 1.40; 95% CI 0.94, 2.10; *p* > 0.05). Among the short-term survivors, women with stage III or IV disease had a 40- to 50-fold increased risk of dying compared to women with stage I disease (stage IV vs. stage I HR 50.05; 95% CI 6.78, 369.57; *p* < 0.05); however, the impact of stage on mortality declined over survival time, as long-term survivors with stage III or IV disease only had up to a 2.76-fold increased risk of dying (stage IV vs. I HR 2.76; 95% CI 1.35, 5.65; *p* < 0.05). Similarly, the presence of residual disease was associated with an increased the risk of death among short-term survivors (HR 2.47; 95% CI 1.40, 4.34; *p* < 0.05), yet this association was attenuated among long-term survivors (HR 1.26; 95% CI 0.83, 1.93; *p* > 0.05).

Breastfeeding, smoking history, and BMI were associated with mortality among long-term but not among short-term survival. In the long-term survival interval, breastfeeding prior to diagnosis significantly decreased the risk of mortality (HR 0.65; 95% CI 0.46, 0.93; *p* < 0.05); however, it was not associated with mortality in the short-term survival interval (HR 1.10; 95% CI 0.85, 1.42; *p* > 0.05). Findings were similar for duration of breastfeeding, as long-term survivors who breastfed for more than 6 months had a lower risk of mortality compared to those who never breastfed (HR 0.61; 95% CI 0.39, 0.95; *p* < 0.05). Among long-term survivors, those with a history of smoking had a higher risk of mortality compared to those who never smoked (HR 1.75; 95% CI 1.27, 2.40; *p* < 0.05). This association was not found among short-term survivors (HR 0.89; 95% CI 0.67, 1.18; *p* > 0.05). Obesity was also associated with a higher risk of mortality compared to a normal BMI among long-term survivors (HR 1.81; 95% CI 1.24, 2.65; *p* < 0.05) but not among short-term survivors (HR 1.31; 95% CI 0.89, 1.94; *p* > 0.05).

There was no significant association between the other exposures related to contraception, parity, menopause, hormone replacement therapy, and fertility with all-cause mortality across the survival time intervals; however, there was a marginally significant association between age at menarche >14 years old and mortality in the intermediate survival interval of 6 to <10 years compared to age at menarche ≤12 years old (HR 1.60; 95% CI 1.01, 2.52; *p* < 0.05).

We also evaluated the associations between the various exposures and ovarian cancer-specific mortality (Appendix A). The findings were similar to all-cause mortality; among long-term survivors, smoking was associated with ovarian cancer-specific mortality to a similar magnitude (HR 1.65; 95%CI 0.94, 2.90), and obesity significantly increased the risk of ovarian cancer-specific mortality (HR 2.15; 95%CI 1.12, 4.10). However, the associations with age at diagnosis and breastfeeding were not statistically significant, which can likely be attributed to the smaller number of ovarian cancer-specific events.

### 3.3. Sensitivity Analyses

In a subset of the population with data on residual disease status following cytoreductive surgery (*n* = 833), additional adjustment for residual disease status did not materially affect the overall findings, including the associations with smoking history and BMI and all-cause mortality (Appendix A). In the subgroup analysis stratifying for tumor stage at diagnosis, the increased risk of all-cause mortality with smoking and obesity among long-term survivors was persistent across all disease stages but was strongest among those diagnosed with late-stage disease (stage III and IV) (Appendix A). Among long-term survivors diagnosed with late-stage ovarian cancer, those with a history of smoking had an HR of 1.98 (95% CI 1.24, 3.15; *p* < 0.05) for all-cause mortality compared to those without a history of smoking. Long-term survivors diagnosed with late-stage ovarian cancer and a BMI categorized as obese had an HR of 2.17 (95% CI 1.27, 3.70) compared to those with a normal BMI. Finally, the main study findings were similar in the analysis limited to HGSC ovarian cancer cases (*n* = 688) (Appendix A).

## 4. Discussion

This study demonstrates the differential association of various exposures and ovarian cancer survival trajectories. Histology, stage, and residual disease following debulking surgery were the most important predictors of overall mortality in all survival trajectories but particularly in the short-term period within 3 years after diagnosis. With respect to modifiable, pre-diagnostic exposures, we observed that breastfeeding, obesity, and smoking did not impact short-term overall survival but significantly impacted long-term overall survival. The probability of ovarian cancer fatality was highest in the 10 years following diagnosis; afterwards; the probability of dying from other causes (i.e., other cancer-related, cardiovascular-related, or respiratory infections) became predominant. Collectively, these findings suggest that pre-diagnostic exposures differentially impact survival time following a diagnosis of ovarian cancer.

To our knowledge, only a few studies have examined the differential impact of exposures on ovarian cancer survival trajectories, the majority of which are based on studies in the U.S. In a population of 35,868 cases from the U.S. SEER database, Peres et al. found that age at diagnosis, stage, marital status, and race/ethnicity were strongly associated with short-term survival; however, the effects waned with long-term survival [11]. We used similar criteria to define short-term survival (<3 years from diagnosis) and long-term survival (≥9 years from diagnosis). In a pooled analysis of 4342 ovarian cancer cases from 22 studies (15 of which were located in the U.S.), Poole et al., reported the differential impacts of age, oral contractive use, and parity on rapidly fatal (<3 years from diagnosis) compared to less aggressive (3+ years from diagnosis) ovarian cancer [9]. Fortner et al. also reported a differential effect of risk factors on tumor aggressiveness (i.e., survival time intervals), with only some risk factors being associated with highly aggressive tumors (<1 year from diagnosis) and others being associated with less aggressive tumors (5+ years from diagnosis) [12].

In the current study, histology, stage, and residual disease following cytoreductive surgery were strongly associated with short-term survival; these strong risk factors for mortality in ovarian cancer patients have been established in many studies [21,22,23]. Those diagnosed with late-stage or serous ovarian cancer were more likely to die within 3 years of diagnosis compared to those diagnosed with stage I or endometrioid disease. The strength of the association declined over survival time. Similar findings were reported by Peres et al., as previously described. In the short-term survival period, distant and regional stage at diagnosis increased the risk of mortality compared to localized disease (HR 8.70; 95% CI 7.72, 9.80 and HR 2.62; 95% CI 2.30, 2.98, respectively). The association declined with long-term survival but remained significant, particularly among high-grade serous carcinomas. Our study found that the presence of residual disease increased the risk of mortality in the short term, yet this association attenuated with long-term survival. To our knowledge, this represents the first analysis of residual disease and ovarian cancer survival trajectories. The current study findings corroborate the existing evidence that achieving maximum cytoreduction maximizes long-term survival probability [24,25].

Pre-diagnostic exposures strongly associated with long-term survival were breastfeeding, smoking, and BMI. History of breastfeeding significantly decreased the risk of mortality among those who survived 10 years or more yet was not significantly associated with short-term survival. Breastfeeding may significantly impact long-term survival, as it is a protective risk factor for overall mortality in the general population [26]. Smoking history and obesity increased the risk of mortality among long-term survivors but had no impact on short-term survival. Again, smoking and obesity are established risk factors for overall mortality in the general population [27,28]; however, other longitudinal studies of pre-diagnostic risk factors in ovarian cancer survivors demonstrate worse survival among those who smoke and have a higher BMI [21]. In contrast, Fortner et al. reported no association with breastfeeding and survival among ovarian cancer survivors [12]. Additionally, the authors reported that obesity and smoking were associated with highly aggressive tumors with survival time intervals <1 year from diagnosis to death (HR 1.93; 95% CI 1.46, 2.56, and HR 1.30; 95% CI 1.07, 1.57, respectively) but not with less aggressive tumors with survival time intervals ≥5 years [12]. Although Fortner et al. reported differential impacts of these modifiable risk factors on ovarian cancer survival, the findings contrast with those in the current study, as they only significantly impacted short-term survival rather than long-term survival. It is possible that these differences can be attributed to the study populations—the majority of the population from Fortner et al.’s study was from the U.S.; therefore, the prevalence and impact of risk factors on survival may differ from this Canadian population [8,12]. The authors used different survival cut-offs, defining highly aggressive tumors as those with a survival time <1 year and long-term survivors as those ≥5 years. Also, it may be that our long follow-up period allowed for the evaluation of long-term impacts of ovarian cancer survival beyond those reported in the Fortner et al. study.

The current study had detailed data on causes of death throughout the survivorship period. The probability of dying from ovarian cancer was highest in the first 10 years after primary diagnosis; thereafter, the probability of dying from other causes predominated. Thus, it is unsurprising that smoking and obesity were associated with long-term mortality from causes other than ovarian cancer. These findings are consistent with other longitudinal studies of ovarian cancer mortality. In a recent population-based study of 6975 women with epithelial ovarian cancer in British Columbia, Canada, Arora et al. similarly reported that ovarian cancer was the predominant cause of death within 10 years of diagnosis, with other causes of death surpassing ovarian cancer after 10 years of survivorship [29]. In a large population-based U.S. study, Dood et al. also characterized a 9-year “high-risk period” of excess ovarian cancer mortality following diagnosis, which was similar to the mortality trends observed in our study [10].

Our study had several strengths, notably, the ability to leverage detailed information from a population-based administrative cancer database, medical chart review, and primary data collection. We included information on pre-diagnostic exposures, pathologic features, treatment characteristics (including residual disease), and updated vital status. The study included a large sample of unselected cases of ovarian cancer with more than 25 years of follow-up. To our knowledge, this is the first analysis of pre-diagnostic exposures on ovarian cancer outcomes by survival trajectories in Canada. We minimized the influence of survivorship bias by conducting a left-truncated survival analysis whereby person-years was initiated at the date of genetic testing.

Our study is not without limitations. Given that this series of ovarian cancer patients only included women who initially consented to and received genetic testing, survivorship bias is possible, i.e., it is possible that some women died prior to being invited to give a blood sample or declined testing due to advanced disease. We sought to reduce the influence of survival bias in our study by using a left-truncated survival analysis (follow-up starting on the date of genetic testing, as previously described [20]). Consequently, we expected only a modest overestimation of 10-year survival in our study (<10%) [20]. Retrospective self-reporting of pre-diagnostic risk factor data, including BMI, smoking history, and breastfeeding, may be subject to measurement and recall bias [30]. We only had information on the extent of residual disease after primary debulking surgery (an important prognostic factor) for a subset (59%) of the patients; however, the study findings remained the same after adjusting for residual disease in sensitivity analysis. Other treatment modalities, such as chemotherapy regimen, were also missing for a large proportion of the cohort, limiting our ability to adjust for this variable in our models. A small proportion (3%) of the population had unknown causes of death.

## 5. Conclusions

In conclusion, we confirmed that clinical factors such as histology, stage, and treatment continue to be important predictors of short- and long-term overall survival; however, we also demonstrated the long-term prognostic role of pre-diagnostic exposures such as breastfeeding, smoking, and BMI. These findings suggest that modifiable risk factors differentially impact ovarian cancer survivorship, which are important potential avenues to improving outcomes. Ovarian cancer mortality remains the primary cause of death in the 10-year period following diagnosis. Women who survive beyond 10 years are encouraged to continue to receive care to manage their risks of dying from other causes.

## Figures and Tables

**Figure 1 cancers-16-00972-f001:**
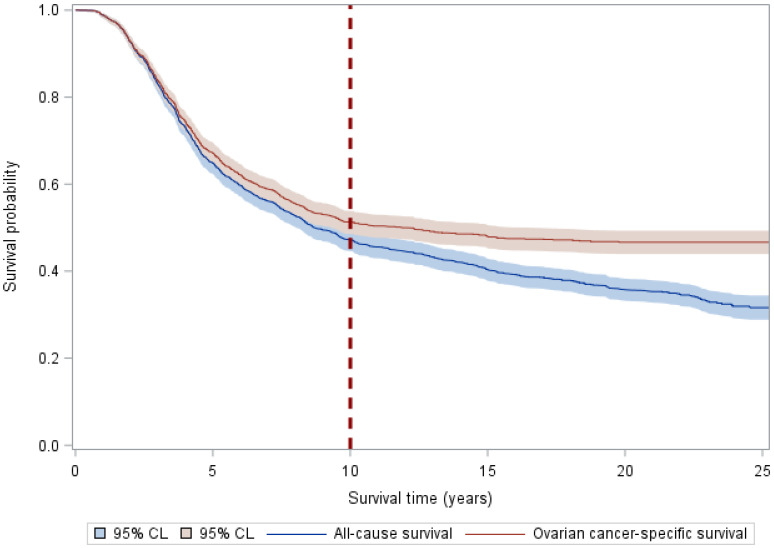
Kaplan–Meier survival curves by all-cause and ovarian cancer-specific mortality. Kaplan–Meier curves plot the survival probability over survival time (years) of all-cause and ovarian cancer-specific mortality. Both all-cause and ovarian cancer-specific survival probabilities decline dramatically over the first 10 years, demonstrating a high-risk period of ovarian cancer death. After 10 years, survival probabilities continue to decline for all-cause mortality but plateau and flatten for ovarian cancer-specific mortality.

**Table 1 cancers-16-00972-t001:** Baseline characteristics of study population overall and by survival time intervals.

Study Characteristic	Overall (*n* = 1421) ^a^	Survival Time			
		<3 Years (*n* = 241)	3–<6 Years (*n* = 330)	6–<10 Years (*n* = 178)	≥10 Years (*n* = 672)
**Follow-up (years), mean (SD)**	11.58 (8.30)	2.09 (0.61)	4.24 (0.80)	7.79 (1.12)	19.59 (4.24)
**Tumor characteristics**					
Age at diagnosis, mean (SD)	57.15 (11.61)	59.44 (11.33)	59.54 (11.24)	58.91 (11.07)	54.70 (11.56)
Histology, *n* (%)					
Serous	783 (55)	185 (77)	241 (73)	117 (66)	240 (36)
Mucinous	115 (8)	6 (2)	5 (2)	6 (3)	98 (15)
Endometrioid	298 (21)	18 (7)	37 (11)	31 (17)	212 (32)
Clear cell	95 (7)	5 (2)	13 (4)	11 (6)	66 (10)
Other	130 (9)	27 (11)	34 (10)	13 (7)	56 (8)
Stage, *n* (%)					
I	254 (18)	1 (1)	7 (2)	12 (7)	234 (35)
II	247 (18)	16 (7)	25 (8)	32 (19)	174 (26)
III	699 (50)	156 (66)	213 (66)	99 (58)	231 (35)
IV	194 (14)	64 (27)	77 (24)	28 (16)	25 (4)
Residual disease, *n* (%)					
No residual disease	300 (36)	14 (9)	39 (18)	29 (29)	218 (61)
Residual disease	533 (64)	148 (91)	173 (82)	70 (71)	142 (39)
**Reproductive/hormonal factors**					
Oral contraceptive use, *n* (%)					
Never	572 (43)	98 (50)	138 (44)	80 (46)	256 (40)
Ever	752 (57)	98 (50)	174 (56)	95 (54)	385 (60)
Duration of oral contraceptive use (years), mean (SD)	2.81 (4.23)	2.72 (4.15)	2.52 (4.37)	2.39 (3.69)	3.08 (4.30)
Estrogen hormone replacement therapy, *n* (%)					
Never	998 (75)	146 (75)	227 (73)	129 (74)	496 (77)
Ever	325 (25)	49 (25)	85 (27)	45 (26)	146 (23)
Endometriosis, *n* (%)					
No	1240 (95)	178 (95)	293 (94)	166 (95)	603 (94)
Yes	72 (5)	10 (5)	18 (6)	8 (5)	36 (6)
Parity, mean (SD) ^b^	2.56 (1.34)	2.75 (1.57)	2.72 (1.35)	2.63 (1.30)	2.40 (1.24)
Age at first birth, mean (SD) ^b^	24.65 (4.82)	24.15 (4.59)	25.04 (5.09)	24.70 (5.27)	24.56 (4.61)
Breastfed, *n* (%) ^b^					
Never	442 (42)	59 (38)	112 (42)	59 (42)	212 (42)
Ever	616 (58)	96 (62)	152 (58)	80 (58)	288 (58)
Duration of breastfeeding (months), mean (SD) ^b^	6.71 (12.62)	6.13 (11.18)	7.67 (13.32)	7.19 (13.34)	6.25 (12.45)
Age at menarche, mean (SD)	12.93 (1.54)	12.90 (1.64)	13.08 (1.57)	12.97 (1.62)	12.87 (1.47)
Age at natural menopause, mean (SD)	49.51 (4.06)	49.19 (4.13)	49.52 (3.91)	49.86 (3.46)	49.51 (4.31)
IUD use, *n* (%)					
Never	1107 (84)	163 (85)	257 (83)	147 (84)	540 (84)
Ever	211 (16)	28 (15)	53 (17)	28 (16)	102 (16)
Number of ovulatory cycles, mean (SD)	387.14 (111.25)	395.21 (114.18)	395.01 (112.82)	405.54 (103.95)	375.69 (110.59)
**Lifestyle factors**					
Smoking, *n* (%)					
Never	676 (51)	106 (54)	156 (50)	84 (48)	330 (51)
Ever	649 (49)	90 (46)	156 (50)	91 (52)	312 (49)
BMI 5 years prior to diagnosis (kg/m^2^), mean (SD)	25.90 (5.49)	26.15 (5.74)	25.90 (5.25)	26.24 (5.75)	25.74 (5.46)
**Family history characteristics**					
*BRCA1/2* mutation, *n* (%)					
No	1244 (88)	210 (87)	273 (83)	144 (81)	617 (92)
Yes	177 (12)	31 (13)	57 (17)	34 (19)	55 (8)
First-degree relatives with history of breast cancer, *n* (%)					
No	1119 (81)	185 (79)	248 (79)	129 (75)	557 (84)
Yes	266 (19)	49 (21)	67 (21)	44 (25)	106 (16)
First-degree relatives with history of ovarian cancer, *n* (%)					
No	1280 (92)	215 (92)	291 (92)	157 (91)	617 (93)
Yes	104 (8)	19 (8)	24 (8)	15 (9)	46 (7)
**Cause of death ^c^**					
Ovarian cancer related	731 (79)	228 (95)	301 (91)	147 (83)	55 (31)
Other cause of death	170 (18)	11 (5)	29 (9)	30 (17)	100 (56)
Unknown	25 (3)	2 (1)	-	1 (1)	22 (12)

^a^ Sample size may vary for each risk factor depending on the availability of the data. ^b^ Among parous women only. ^c^ Among women who died only.

**Table 2 cancers-16-00972-t002:** HR and 95% CI of risk factors and all-cause mortality by survival time intervals ^a^.

Risk Factors	N	<3 years		3–<6 Years		6–<10 Years		≥10 Years	
		Cases	HR (95% CI)	Cases	HR (95% CI)	Cases	HR (95% CI)	Cases	HR (95% CI)
**Tumor characteristics**									
Age at diagnosis, *n* (%)									
<50 years	390	50	0.94 (0.65, 1.36)	67	0.95 (0.69, 1.30)	37	0.81 (0.53, 1.24)	26	0.43 (0.27, 0.69) ^†^
50 to <60 years	410	65	1.00 (ref)	93	1.00 (ref)	51	1.00 (ref)	49	1.00 (ref)
60 to <70 years	348	69	1.23 (0.88, 1.72)	85	1.17 (0.87, 1.58)	49	1.35 (0.91, 2.00)	60	2.08 (1.42, 3.05) ^†^
≥70 years	246	53	1.20 (0.83, 1.72)	77	1.55 (1.15, 2.11) ^†^	34	1.55 (1.00, 2.40) ^†^	42	2.93 (1.93, 4.46) ^‡^
Histology									
Serous	758	181	1.97 (1.20, 3.22) ^†^	233	1.70 (1.91, 2.44) ^†^	112	1.49 (0.98, 2.78)	83	1.40 (0.94, 2.10)
Mucinous	115	6	1.33 (0.52, 3.37)	5	0.46 (0.18, 1.16)	6	0.49 (0.21, 1.18)	27	1.34 (0.83, 2.17)
Endometrioid	298	18	1.00 (ref)	37	1.00 (ref)	31	1.00 (ref)	43	1.00 (ref)
Clear cell	94	5	1.04 (0.39, 2.83)	13	1.50 (0.79, 2.83)	10	1.10 (0.54, 2.24)	12	0.90 (0.47, 1.72)
Other	129	27	1.89 (1.03, 3.45) ^†^	34	1.50 (0.93, 2.41)	12	0.72 (0.36, 1.43)	12	0.72 (0.37, 1.39)
Stage									
I	254	1	1.00 (ref)	7	1.00 (ref)	12	1.00 (ref)	47	1.00 (ref)
II	247	16	13.55 (1.79, 102.68) ^†^	25	3.28 (1.41, 7.62) ^†^	32	2.73 (1.39, 5.35) ^†^	43	1.17 (0.76, 1.80)
III	699	156	40.26 (5.53, 293.29) ^†^	213	11.97 (5.50, 26.07) ^‡^	99	5.15 (2.72, 9.75) ^‡^	77	1.65 (1.10, 2.49) ^†^
IV	194	64	50.05 (6.78, 369.57) ^†^	77	20.33 (9.10, 45.45) ^‡^	28	11.81 (5.77, 24.21) ^‡^	10	2.76 (1.35, 5.65) ^†^
Residual disease									
No	300	14	1.00 (ref)	39	1.00 (ref)	29	1.00 (ref)	55	1.00 (ref)
Yes	533	148	2.47 (1.40, 4.34) ^†^	173	2.03 (1.41,2.92) ^†^	70	1.79 (1.12, 2.84) ^†^	54	1.26 (0.83, 1.93)
**Reproductive/hormonal factors**									
Oral contraceptive use									
Never	568	97	1.00 (ref)	138	1.00 (ref)	79	1.00 (ref)	81	1.00 (ref)
Ever	732	97	0.85 (0.62, 1.17)	166	1.10 (0.86, 1.42)	90	0.97 (0.69, 1.36)	81	0.81 (0.59, 1.11)
Duration of oral contraceptive use (years)									
0	570	84	1.00 (ref)	142	1.00 (ref)	75	1.00 (ref)	81	1.00 (ref)
1–5	307	42	1.08 (0.74, 1.60)	74	1.10 (0.82, 1.47)	41	1.15 (0.77, 1.70)	31	1.02 (0.66, 1.56)
>5	219	30	1.04 (0.67, 1.63)	42	0.88 (0.61, 1.26)	25	0.82 (0.51, 1.33)	23	0.99 (0.60, 1.62)
Estrogen hormone replacement therapy									
Never	977	145	1.00 (ref)	220	1.00 (ref)	124	1.00 (ref)	115	1.00 (ref)
Ever	322	48	0.94 (0.67, 1.31)	84	0.87 (0.68, 1.15)	44	0.79 (0.55, 1.23)	47	1.07 (0.75, 1.53)
Endometriosis									
No	1219	177	1.00 (ref)	287	1.00 (ref)	160	1.00 (ref)	157	1.00 (ref)
Yes	70	10	1.19 (0.62, 2.27)	16	1.18 (0.71, 1.98)	8	1.04 (0.50, 2.15)	4	0.64, (0.23, 1.77)
Parity, never/ever									
Nulliparous	218	36	1.00 (ref)	38	1.00 (ref)	29	1.00 (ref)	26	1.00 (ref)
Parous	1084	159	0.78 (0.54, 1.12)	266	1.15 (0.81, 1.62)	140	0.92 (0.61, 1.39)	135	0.76 (0.49, 1.19)
Parity ^b^									
1	190	21	1.00 (ref)	36	1.00 (ref)	26	1.00 (ref)	23	1.00 (ref)
2	407	63	1.33 (0.81, 2.19)	99	1.26 (0.86, 1.85)	46	0.96 (0.59, 1.56)	50	1.12 (0.68, 1.84)
>3	440	69	1.17 (0.71, 1.93)	122	1.30 (0.89, 1.90)	61	1.13 (0.70, 1.82)	58	1.00 (0.61, 1.64)
Age at first birth ^b^									
<20	117	20	1.00 (ref)	26	1.00 (ref)	16	1.00 (ref)	17	1.00 (ref)
20–30	685	92	0.79 (0.48, 1.28)	167	1.08 (0.72, 1.64)	85	0.77 (0.45, 1.32)	85	0.67 (0.39, 1.14)
>30	107	12	0.55 (0.27, 1.13)	35	1.52 (0.91, 2.53)	15	1.10 (0.54, 2.23)	12	0.99 (0.47, 2.11)
Breastfed ^b^									
Never	434	58	1.00 (ref)	108	1.00 (ref)	57	1.00 (ref)	71	1.00 (ref)
Ever	603	95	1.28 (0.92, 1.79)	149	1.10 (0.85, 1.42)	76	1.07 (0.75, 1.51)	60	0.65 (0.46, 0.93) ^†^
Duration of breastfeeding (months) ^b^									
0	427	58	1.00 (ref)	105	1.00 (ref)	55	1.00 (ref)	70	1.00 (ref)
1–6	286	46	1.34 (0.90, 2.00)	61	0.96 (0.70, 1.33)	37	1.09 (0.71, 1.66)	32	0.70 (0.46, 1.08)
>6	313	46	1.14 (0.77, 1.69)	88	1.23 (0.92, 1.64)	39	1.11 (0.74, 1.69)	28	0.61 (0.39, 0.95) ^†^
Age at menarche (years)									
≤12	489	85	1.00 (ref)	100	1.00 (ref)	60	1.00 (ref)	62	1.00 (ref)
13	409	47	0.71 (0.49, 1.01)	100	1.19 (0.90, 1.57)	51	1.00 (0.69, 1.47)	51	0.75 (0.51, 1.10)
14	221	33	0.71 (0.47, 1.07)	53	1.10 (0.78, 1.54)	30	1.11 (0.71, 1.74)	25	0.72 (0.45, 1.16)
>14	178	29	0.81 (0.53, 1.24)	49	1.33 (0.94, 1.89)	28	1.60 (1.01, 2.52) ^†^	23	1.07 (0.65, 1.73)
Age at natural menopause									
≤47	175	34	1.00 (ref)	46	1.00 (ref)	22	1.00 (ref)	28	1.00 (ref)
>47–50	228	31	0.82 (0.50, 1.34)	53	0.77 (0.52, 1.16)	39	1.14 (0.67, 1.98)	38	1.08 (0.65, 1.82)
>50–52	147	20	0.86 (0.49, 1.50)	46	1.15 (0.75, 1.75)	14	0.86 (4.3, 1.74)	19	0.79 (0.43, 1.45)
>52	142	20	0.70 (0.40, 1.24)	37	0.83 (0.53, 1.29)	20	0.96 (0.51, 1.82)	25	1.10 (0.63, 1.93)
Menopausal status at diagnosis									
Premenopausal	370	48	1.00 (ref)	64	1.00 (ref)	40	1.00 (ref)	30	1.00 (ref)
Postmenopausal	907	143	1.00 (0.59, 1.69)	235	1.28 (0.81, 2.02)	129	1.10 (0.61, 1.97)	132	1.12 (0.61, 2.07)
IUD use									
Never	1090	162	1.00 (ref)	251	1.00 (ref)	143	1.00 (ref)	139	1.00 (ref)
Ever	206	28	0.89 (0.59, 1.35)	52	1.07 (0.79, 1.46)	26	0.99 (0.64, 1.51)	23	1.12 (0.71, 1.77)
Number of ovulatory cycles									
≤322.28	228	29	1.00 (ref)	54	1.00 (ref)	20	1.00 (ref)	23	1.00 (ref)
>322.28–389.52	230	37	1.30 (0.78, 2.17)	52	0.81 (0.54, 1.20)	36	1.34 (0.75, 2.38)	31	1.14 (0.63, 2.07)
>389.52–439.42	225	30	1.05 (0.59, 1.89)	61	0.90 (0.60, 1.34)	28	1.08 (0.58, 2.04)	32	1.06 (0.57, 1.99)
>439.42	225	33	1.20 (0.67, 2.16)	58	0.94 (0.61, 1.43)	31	1.20 (0.63, 2.30)	33	0.99 (0.53, 1.86)
**Lifestyle factors**									
Smoking, never/ever									
Never	661	104	1.00 (ref)	151	1.00 (ref)	80	1.00 (ref)	64	1.00 (ref)
Ever	640	90	0.89 (0.67, 1.18)	153	0.97 (0.77, 1.22)	89	1.12 (0.83, 1.52)	98	1.75 (1.27, 2.40) ^†^
BMI 5 years prior to diagnosis (kg/m^2^)									
Underweight	20	6	1.69 (0.73, 3.90)	2	0.46 (0.11, 1.85)	3	0.99 (0.31, 3.17)	1	0.47 (0.07, 3.38)
Normal	660	94	1.00 (ref)	155	1.00 (ref)	81	1.00 (ref)	81	1.00 (ref)
Overweight	388	56	1.12 (0.80, 1.56)	95	1.04 (0.80, 1.34)	58	1.32 (0.94, 1.86)	39	0.84 (0.57, 1.24)
Obese	226	35	1.31 (0.89, 1.94)	51	1.09 (0.80, 1.50)	27	1.09 (0.70, 1.70)	41	1.81 (1.24, 2.65) ^†^

^a^ All risk factors were adjusted for age at diagnosis, histology, and stage. ^b^ Among parous women only. ^†^
*p* < 0.05, ^‡^
*p* < 0.001.

## Data Availability

The data that support the findings of this study are available on request from the corresponding author. The data are not publicly available due to privacy or ethical restrictions.

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
