# Peer review of "Impact of Pre-Diagnostic Risk Factors on Short- and Long-Term Ovarian Cancer Survival Trajectories: A Longitudinal Observational Study"

_cancers, 2024, doi:10.3390/cancers16050972_

Round 1

Reviewer 1 Report

Comments and Suggestions for Authors

I had the opportunity to review the manuscript titled "Impact of pre-diagnostic risk factors on short- and long-term ovarian cancer survival trajectories: a longitudinal observational study" by Kim et al., submitted to Cancers. Overall, I found the study to be well-conducted and the findings to be of significant interest. However, I would like to suggest some minor revisions and seek clarifications on certain aspects of the manuscript before its publication.

1. The authors state that they "previously demonstrated that pre-diagnostic lifestyle factors, notably obesity and smoking, were significant predictors of death among women diagnosed with epithelial ovarian cancer in Ontario (Canada) after accounting for established clinical and pathologic prognostic factors." I recommend that the authors provide more specific information about their previous study, such as the time frame, stages evaluated (if applicable), and any other relevant data. This additional context will enhance the understanding of the impact of pre-diagnostic lifestyle factors.

2. The choice to focus exclusively on epithelial ovarian cancer is well-justified due to its prevalence. However, the authors should explicitly address why other ovarian cancer types were not included. An explanation of this decision will contribute to a more comprehensive understanding of the study's scope.

3. The recruitment periods between January 1995 and December 1999 and between January 2002 and December 2004 are specified, but the rationale for these specific time gaps and the exclusion of the years in between should be elucidated for transparency. A brief discussion of why these periods were chosen will enhance the study's methodological clarity.

4. Regarding genetic testing, the authors state that the study's primary focus is to investigate the association between BRCA1 and BRCA2 mutations and ovarian cancer. If this is the case, the authors should clarify whether they evaluated the genotype of these mutations in their study. Additionally, consideration of other potential genetic markers beyond BRCA1 and BRCA2 may be discussed.

5. The authors mention obtaining data from various sources, and it would be beneficial for readers to understand how the data were standardized and normalized. Providing details on these processes will enhance the robustness of the data analysis.

6. While the study mentions smoking and BMI as lifestyle-associated risk factors, the authors might consider expanding the discussion to include other factors such as diet, Talcum Powder Use, and inherited conditions like Lynch syndrome. A brief exploration of these factors will contribute to a more comprehensive analysis.

7. The finding related to the subgroup analysis on tumor stage at diagnosis, indicating a persistent increased risk of all-cause mortality with smoking and obesity among long-term survivors, is particularly noteworthy. I recommend considering the inclusion of this finding in the main manuscript rather than as supplementary material, and further discussion of its implications would enhance the manuscript's impact.

Thank you for considering these suggestions. I look forward to the revised manuscript and the potential publication of this important work in Cancers.

Sincerely,

Comments on the Quality of English Language

There are some minor grammatical errors, especially in the discussion which should be addressed. 

Reviewer 2 Report

Comments and Suggestions for Authors

The manuscript is well-written and all the sections are detailed. 

1. In line 87-88, the years of data collection have a gap of 2 years. Do the authors have a specific reason for restricting the data to those specific years? 

2. Is the formula, provided on lines 110-112 for cumulative ovulatory cycles derived or from existing literature. Add reference accordingly. 

3. Were any other treatment modalities also assessed, as it might also have an impact on the survival. 

4. In the discussion lines 356-357, the self-reported data with retrospective data collection would also lead to recall bias. 

Reviewer 3 Report

Comments and Suggestions for Authors

The authors have provided a well-written manuscript on the impact of pre-diagnostic risk factors on ovarian cancer survival. The authors followed 1421 women diagnosed with ovarian cancer after genetic testing offering and correlated some tumour characteristics and pre-diagnostic factors with the overall survivorship of these patients. Sound statistical methods have been applied, and the methods and results are presented. Nonetheless, there are a few things that the authors should clarify within the manuscript and perhaps adjust the discussions and results sections accordingly.

1. It is unclear why only the patients who accepted genetic counselling and consecutive testing were included in this study. This is not a significant problem, but the authors can further elucidate this selection of patients.

2. On numerous occasions, the authors name the terms rapidly fatal and long-term disease (as a binary classification) but fail to define exact numbers for these categories. I acknowledge the survivorship categories split into 4 (<3, >3<6, >6<10 and >10) that were used for the results section, but I do not seem to recall any more information regarding the rapidly fatal and long-term survival binary classification of the disease and its relevance further down the manuscript, beyond the summary/abstract and introduction sections. Please clarify.

3. Sound statistical methods have been applied, but unremarkable results were yielded. Regarding the tumour characteristics, age, histology, stage and residual disease correlated with overall survivorship, which is unsurprising and well-established for decades (du Bois A, Reuss A, Pujade-Lauraine E, Harter P, Ray-Coquard I, Pfisterer J. Role of surgical outcome as prognostic factor in advanced epithelial ovarian cancer: a combined exploratory analysis of 3 prospectively randomized phase 3 multicenter trials: by the Arbeitsgemeinschaft Gynaekologische Onkologie Studiengruppe Ovarialkarzinom (AGO-OVAR) and the Groupe d'Investigateurs Nationaux Pour les Etudes des Cancers de l'Ovaire (GINECO). Cancer. 2009 Mar 15;115(6):1234-44. doi: 10.1002/cncr.24149. PMID: 19189349.). Lesser investigated factors, such as reproductive/hormonal and lifestyle factors grouped as pre-diagnostic risk factors have also been investigated in the cohort of patients. Of these, it appears that breastfeeding, smoking and obese-values of BMI scores correlated to long-term survivorship. These findings have a few issues: a. Results evaluate all-cause mortality and not ovarian cancer mortality b. The Number of long-term deaths (> 10 years) is very small compared to early deaths c. Most causes of death occurring at this time frame are not ovarian-cancer-related deaths (55 out of 731) but rather other causes (100 out of 170) or unknown (22 out of 25), which are probably not ovarian cancer-related either. This is very important because most ovarian cancer patients (as most cases are diagnosed at stages III and IV and the majority have an aggressive epithelial histology) do not survive long term. Correlating pre-diagnosis risk factors with overall survival (all-cause mortality) in this category is extremely sensitive because you will probably correlate risk factors to overall mortality and not to ovarian cancer-specific mortality. It is obvious that smoking and obesity will impact long-term survivorship in any patient, regardless of their ovarian cancer trajectory. Since the majority of deaths in this category were not ovarian cancer-related, probably the risk factors were neither. Your results exhibit general risk factors for all-cause mortality at ten years. However, they happen to be in an ovarian cancer patient cohort that was lucky enough to survive so long. This is supported by the more important findings only displayed in the supplementary material as ovarian-cancer-specific mortality. Naturally, results exhibit a lack of statistically significant correlations between such risk factors and long-term survivorship, as expected. There is no flaw in the statistical methods applied or any part of the results, but rather, a major flaw in how the results are presented and discussed. Various affirmations such as: „Findings were similar to all-cause mortality” or „With respect to modifiable, pre-diagnostic exposures, we observed that breastfeeding, obesity, and smoking did not impact short-term survival, but significantly impacted long-term survival” are not false but inherently misleading. You can plainly see that 20% of non-ovarian cancer deaths after 10 years are cancer-related, and 15% are cardiovascular events. So, it is no surprise that smoking and obesity are correlated here.

4. Authors make some points in the discussion setting: „it is possible some women had died prior to be invited to give a blood 351 sample or declined testing due to advanced disease” and „Consequently, we expect only a modest overestimation of 10-year survival in our study (<10%)" which i believe is very optimistic. Given that more than 35% of the population were stages I/II of the disease and only about 50% of histologies were aggressive epithelial serous histologies, 47% of the cohort survived >10 years. This is very impressive, considering the patients were followed at a time when maintenance therapies such as bevacizumab and newer PAPR-I molecules were unheard of. Either that, or we are indeed looking at a bias in selecting patients that is seriously underestimated.

Overall, this is a well-written manuscript, but the results are misleading or presented in a very favourable manner. I repeat, I firmly believe that your cohort has some selection bias and that the findings only correlate general risk factors to all-cause mortality in a highly selected lucky ovarian cancer patients cohort in which half of the women survived more than 10 years. I would rather move the unremarkable findings regarding ovarian-cancer-specific deaths from the supplementary material into the main text and adjust the conclusions accordingly.

Comments on the Quality of English Language

The article is very well-written. English language is not a problem.

Reviewer 4 Report

Comments and Suggestions for Authors

The authors should be congratulated for the interesting topic discussed. 

1.     Methods and methodology are robust.

2.     Results and conclusions are well presented.

3.     Tables and graphics are clearly described.

The study has sufficient merit to be considered for publication, although major revisions are required because it is worth mentioning more current progress done in scientific and clinical fields when it comes to predicting tumoral recurrence or treatment unresponsiveness and a better insight into tumoral risk factors, in general. I suggest providing more detailed information about it. These papers can offer a valid solution (https://doi.org/10.1016/j.urolonc.2022.05.016https://doi.org/10.1016/j.clgc.2021.12.005, https://doi.org/10.3390%2Fjpm13030512). A lecture is suggested.

Comments on the Quality of English Language

Minor editing.

Round 2

Reviewer 4 Report

Comments and Suggestions for Authors

Authors answered all comments and suggestions.

Comments on the Quality of English Language

Minor editing.